# Zika Virus Infection in a Cohort of Pregnant Women with Exanthematic Disease in Manaus, Brazilian Amazon

**DOI:** 10.3390/v12121362

**Published:** 2020-11-28

**Authors:** Elijane de Fátima Redivo, Camila Bôtto Menezes, Márcia da Costa Castilho, Marianna Brock, Evela da Silva Magno, Maria das Graças Gomes Saraiva, Salete Sara Alvarez Fernandes, Anny Beatriz Costa Antony de Andrade, Maria das Graças Costa Alecrim, Flor Ernestina Martinez-Espinosa

**Affiliations:** 1Postgraduate Program in Tropical Medicine, University of Amazonas State, Manaus 69040-000, Brazil; elijaneredivo@gmail.com (E.d.F.R.); camila.chabm@gmail.com (C.B.M.); saletesarafernandes@gmail.com (S.S.A.F.); galecrim.br@gmail.com (M.d.G.C.A.); 2Department of Medicine, School of Health Sciences, University of Amazonas State, Manaus 69065-000, Brazil; mariannabrock@hotmail.com; 3Department of Malaria, Tropical Medicine Foundation Dr. Heitor Vieira Dourado, Manaus 69040-000, Brazil; 4Department of Virology, Tropical Medicine Foundation Dr. Heitor Vieira Dourado, Manaus 69040-000, Brazil; mcastilho@fmt.am.gov.br; 5Department of Epidemiology and Public Health, Tropical Medicine Foundation Dr. Heitor Vieira Dourado, Manaus 69040-000, Brazil; evelamagno_2011@hotmail.com (E.d.S.M.); gracag.saraiva@gmail.com (M.d.G.G.S.); 6Postgraduate Program in Living Conditions and Health Situations in the Amazon PPGVIDA, Leônidas & Maria Deane Institute, Manaus 69057-070, Amazonas, Brazil; antony.beatriz@gmail.com; 7Laboratory of Territory, Environment, Health and Sustainability, Leônidas & Maria Deane Institute, Manaus 69057-070, Fiocruz Amazonia, Brazil

**Keywords:** Amazonian region, exanthematic disease in pregnancy, TORCH Syndrome, ZIKV infection, miscarriage, stillbirth, microcephaly, preterm delivery, low birth weight

## Abstract

The epidemic transmission of Zika virus (ZIKV) in Brazil has been identified as a cause of microcephaly and other neurological malformations in the babies of ZIKV-infected women. The frequency of adverse outcomes of Zika virus infection (ZIKVi) in pregnancy differs depending on the characteristics of exposure to infection, the time of recruitment of research participants, and the outcomes to be observed. This study provides a descriptive analysis—from the onset of symptoms to delivery—of a cohort registered as having maternal ZIKVi in pregnancy, from November 2015 to December 2016. Suspected cases were registered at a referral center for infectious and tropical diseases in Manaus, in the Amazonian region of Brazil. Of 834 women notified, 762 women with confirmed pregnancies were enrolled. Reverse-transcriptase polymerase chain reaction (RT-PCR) confirmed ZIKVi in 42.3% of the cohort. In 35.2% of the cohort, ZIKV was the sole infection identified. Severe adverse pregnancy outcomes (miscarriage, stillbirth, or microcephaly) were observed in both RT-PCR ZIKV-positive (5.0%) and ZIKV-negative (1.8%) cases (RR 3.1; 95% IC 1.4–7.3; *p* < 0.05), especially during the first trimester of pregnancy (RR 6.2, 95% IC 2.3–16.5; *p* < 0.001). Although other infectious rash diseases were observed in the pregnant women in the study, having confirmed maternal ZIKVi was the most important risk factor for serious adverse pregnancy events.

## 1. Introduction

The association of Zika virus infection (ZIKVi) in pregnant women with microcephaly in newborns has raised the concern of health authorities during the epidemic of this arbovirus. Since 2015, several studies focusing mainly on maternal infection effects in children have been published [1,2,3,4,5,6]. The risk of maternal ZIKVi for pregnancy, fetus, and child may vary, depending on the study design and maternal exposure to the infection. Therefore, assessing the problem in different geographic regions and different epidemiological contexts contributes to the understanding of the issue. This study describes the evolution of pregnancy—from the onset of symptoms to delivery—in pregnant women with exanthematic disease, according to the results of ZIKV reverse-transcriptase polymerase chain reaction (RT-PCR), and other infectious agents causing exanthema during a period of intense transmission of ZIKV in the Brazilian Amazon, showing the importance of follow-up for pregnant women, despite the result of the ZIKV RT-PCR.

## 2. Materials and Methods

This study describes a cohort of notified-as-suspected cases of ZIKVi in pregnancy. The patients in the cohort were followed up from the onset of symptoms until delivery for 58 weeks, which lasted from the 47th epidemiological week of 2015 to the last epidemiological week of 2016. Patients were attended to at the Tropical Medicine Foundation Dr Heitor Vieira Dourado (FMT-HVD), a referral center for infectious diseases in Manaus, which is in the state of Amazonas, Brazil. FMT-HVD is neither an antenatal clinic nor a maternity center. Living in a region where mosquito-borne transmission of ZIKV has been reported, the entire population enrolled was considered to have a history of exposure.

### 2.1. Study Population

The disease burden in the city of Manaus was estimated based on the reported and confirmed cases of ZIKVi in pregnant women by the Health Surveillance Foundation (FVS) per thousand live births in the same period municipality, according to the National System of Live Births (SINASC). During the period of epidemic transmission, pregnant women were regarded as a high-priority group to diagnose ZIKVi quickly and with high specificity using reverse transcription-polymerase chain reaction (RT-PCR). Women who, when seeking care at the FMT-HVD, declared themselves pregnant and had a rash accompanied by two or more symptoms suggestive of ZIKVi were eligible to enter the study. Blood and urine samples were collected from symptomatic women of childbearing age, who also received ambulatory care.

### 2.2. Definitions

Pregnancy was confirmed using a beta human chorionic gonadotropin (HCG) test, ultrasonography (USG), or clinical examination. The stage of pregnancy was calculated from the first day of the last regular menstruation period, by USG performed in the first trimester of pregnancy, or otherwise determined from the maturity of the newborn baby. The first trimester was defined as lasting from 0 to 13.3 weeks of amenorrhea, the second trimester was defined as lasting from 13.4 to 26.6 weeks of amenorrhea, and the third trimester was defined as occurring after that.

A maternal infection case was defined as suspected ZIKVi if the pregnant patient had a macular or papular rash with two or more other symptoms, such as fever, conjunctival hyperemia without secretion, pruritus, or arthralgia of the hands or feet. A case was defined as confirmed maternal ZIKVi if a rRT-PCR test for ZIKV showed positive results. The definitions of suspected and confirmed maternal ZIKVi cases followed those used by the Ministry of Health of Brazil [7]. RT-PCR results were classified into five categories: ZIKV RT-PCR positive cases, ZIKV RT-PCR negative cases, ZIKV RT-PCR with indeterminate results, ZIKV RT-PCR that was “under analysis” (analysis not completed), and ZIKV RT-PCR not tested. All cases were enrolled and followed until delivery. Any interruption of pregnancy before the 22nd week of pregnancy with fetal loss was recorded as a miscarriage. Stillbirth was defined as a fetal loss after the 22nd week of pregnancy. Premature birth was defined as the interruption of pregnancy with a live newborn occurring before week 37 of pregnancy. Low birth weight was defined as a weight of less than 2500 g at birth. Microcephaly was defined as a head circumference more than two standard deviations (SDs) below the expected mean for the same sex and gestational age. The pregnancy outcome was defined as severe (SPO) if miscarriage, stillbirth, or microcephaly occurred. The pregnancy outcome was defined as moderate (MPO) if low birth weight and/or pre-term delivery occurred. If none of these events were identified, the pregnancy was defined as having no adverse effects (NAPO). Other adverse effects of exposure inside the uterus may appear or manifest at different times throughout the child’s life, requiring prolonged follow-up; therefore, detailed postnatal evaluation of the infant and their follow-up from birth and during the first few years of life will be reported in a future study.

### 2.3. Patient Care

FMT-HVD is a referral center for tropical and infectious diseases in the state of Amazonas in Brazil. For this reason, it is one of the most sought-out institutions by members of the population, especially in epidemic situations such as ZIKV transmission. FMT-HVD is also one of the most important education and research institutions in tropical medicine and infectious diseases in the Brazilian Amazonian region.

Due to the known associations of maternal ZIKVi with teratogenesis and malformations in the fetus, pregnant women sought out diagnostic confirmation and procedures that they should follow. The patients were received by the emergency services room and were subsequently evaluated by the epidemiological surveillance service, which referred patients to outpatient care with the research team. In most cases, the pregnant women underwent outpatient screening and were attended to by the epidemiological surveillance service and outpatient care.

### 2.4. Laboratory Testing

Sample collection mostly occurred when the pregnant women received their first care at the institution, although outpatient care was provided at various times, ranging from the same day to within a week of receipt of first care. During the first or a subsequent instance of outpatient care, all of the patient’s demographic and clinical data were recorded on a form and a new laboratory test (for TORCH) was required by an infectious disease specialist belonging to our research team; subsequent contact occurred four or five weeks later. As the evaluation at FMT-HVD was not a replacement for antenatal care, all pregnant women were referred to a high-risk prenatal service, created for this purpose by the prefecture of Manaus.

Confirmatory tests for ZIKV during acute gestational infection [8] were performed at the Central Laboratory of Public Health (LACEN) in Manaus using the reverse transcriptase reaction, followed by real-time polymerase chain reaction (RT-qPCR) in serum and urine, as well as immunoenzymatic tests for Chikungunya. Collection of serum samples within the first five days of symptom onset was strongly encouraged, as was the collection of urine samples within the first eight days of symptom onset. Tests for Dengue virus and Parvovirus B19 virus infection were performed by the Virology Laboratory at the FMT-HVD, using Dengue Vírus IgM Capture and Parvovirus B19 IgM kits (DxSelect^TM^ Focus Diagnostics, Cypress, CA, USA). The detection of etiological agents of TORCH Syndrome was mainly performed at the Clinical Analysis Laboratory of the FMT-HVD; however, if the patient had recent test results from a public- or private-network laboratory, new tests were not performed.

### 2.5. Ethics

Written informed consent was obtained from each patient for the study, with their signature of the Informed Consent Form (ICF), under the approval of the ethics committee obtained from the Tropical Medicine Foundation Dr Heitor Vieira Dourado (FMT-HVD) Ethics Review Board under the Certificate of Presentation of Ethical Appreciation (CAAE 60168216.2.0000.0005) approval number 1′806.030 in 4 November 2016.

### 2.6. Statistical Analysis

All data were stored in an Excel spreadsheet (Microsoft, Redmond, WA, USA) and analyzed using the Epi Info 7 (Centers of Disease Control and Prevention, Atlanta, GA, USA) and Minitab 17 (Minitab, State College, PA, USA) software. Pregnant women who were RT-PCR positive for ZIKV were compared with those whose RT-PCR for ZIKV was not positive using Fisher’s exact test (two-sided). Chi-square tests (McNemar’s tests) were used to compare discrete data. Normally distributed data were analyzed with Student’s *t*-test or ANOVA to compare continuous data. *p* values less than 0.05 were regarded as statistically significant.

## 3. Results

### 3.1. Case Incidence in Manaus

The Health Surveillance Foundation (FVS), which is responsible for the notification of diseases in the municipality of Manaus, notified 1286 suspected cases of maternal ZIKVi in pregnant women, confirming 500 of these, during the period of this study. During the same time, 47,791 births were reported by the Information System of Natality (SINASC). It was estimated that the disease burden was 10.4 cases per 1000 liveborn babies in Manaus.

Our ambulatory clinic for infectious diseases in pregnancy at the FMT-HVD attended to 834 of the suspected cases (64.9%) and 322 (67.8%) of the confirmed cases in Manaus. All women who were identified as having a suspected case of maternal ZIKVi at the FMT-HVD appear in Figure 1. Blood and urine samples were collected from the patients, and the presence of positive RT-PCR ZIKV findings in either sample was used to confirm maternal ZIKVi. During the recruitment process, 12 women who did not have their pregnancy confirmed were excluded; another 60 women had a loss of follow-up before birth, while 762 pregnant women with the rash disease were included in the study.

### 3.2. Temporal Distribution of Cases

The temporal distribution of the inclusion of research participants is shown in Figure 2. During the 58 weeks, 762 maternal ZIKVi in pregnancy suspected cases were followed by the FMT-HVD.

### 3.3. Clinical Courses

#### 3.3.1. Case Characteristics and Presentation

The characteristics of the included women are summarized in Table 1. The women were most likely to be younger adults (aged 18 to 29 years), to be married or in a stable union, to have a high school education, to have had at least one previous pregnancy, and to be in the first half of pregnancy (Appendix A). Symptom frequencies were also analyzed, according to the RT-PCR ZIKV test results, as shown in Table 2. When the symptoms of all the isolated infections were analyzed, rash and pruritus were found to be the most frequent as seen in Table 2 and Appendix A.

#### 3.3.2. Testing for Infections that Produce TORCH Syndrome

Maternal ZIKVi (42.3%) was the most prevalent of the detected infections, being the only infection detected in 35.3% of patients. However, cases of maternal dengue virus (4.7%), maternal syphilis (0.8%), maternal HIV (0.9%), maternal herpes 1 and 2 (6.2%), and maternal parvovirus (3.3%) infection were also detected, as well as some additional infection types and cases of multiple simultaneous infections. In 49.7% of the patients, no infection was found (Figure 3). The time that had elapsed between the onset of symptoms and sample collection was lower in the RT-PCR positive patients than for the RT-PCR negative patients (mean ± SD: 2.7 ± 2.9 vs. 2.9 ± 4.3 days, *p* < 0.001).

#### 3.3.3. Pregnancy Outcomes

In the patient cohort, the most severe adverse outcomes of pregnancy were miscarriage (10 cases, 1.3%), stillbirth (7 cases, 0.9%), and microcephaly (7 cases, 0.9%), as shown in Figure 1. The onset of symptoms in the first trimester of pregnancy had the greatest statistical significance as a risk factor for severe adverse outcomes, compared to the other two trimesters (RR, 8.1; 95% CI, 3.4–19.2). When the first trimester was compared to the second trimester, the risk of serious adverse pregnancy events was less than the risk identified when comparing the onset of symptoms to the third trimester (RR, 6.2; 95% CI, 2.3–16.5; *p* < 0.001 and RR, 12.8; 95% CI, 3.0–54.0; *p* < 0.001, respectively). No statistically significant association was identified between parity or sex of the newborn and serious adverse pregnancy events.

Although severe adverse outcomes were most common among RT-PCR ZIKV-positive patients (RR, 3.1; 95% CI, 1.4–7.3; *p* < 0.05), especially during the first trimester of pregnancy (RR, 6.2; 95% CI, 2.3–16.5; *p* < 0.001), they were also observed in RT-PCR non-positive pregnant women (Table 3 and Table 4). Both in cases of miscarriage and stillbirths, as well as ZIKV-negative microcephaly, no other etiological agent was evidenced, except for one miscarriage case with positive results for toxoplasmosis and parvovirus (Table 3).

## 4. Discussion

In this study, we assessed the evolution of pregnancy–from the onset of symptoms to the end of pregnancy–in a large population of pregnant women with rash disease during a period of intense local transmission of the Zika virus in the Brazilian Amazon. Although less than half of the pregnant women had their infection confirmed in the laboratory, adverse events in the fetus and child were observed in the entire population studied. Other diseases that cause similar conditions in pregnant women and fetal infections by vertical transmission were also studied and were not identified in cases with severe adverse pregnancy outcome, except in a fetal loss associated with Toxoplasmosis and Parvovirus B19-coinfection. Serious adverse outcomes in the fetus, such as miscarriage, stillbirth, or microcephaly, were more likely to occur in pregnant women who had confirmed Zika virus infection, especially if symptoms started in the first trimester of pregnancy.

The most frequent question that a health professional had to answer to pregnant women during the Zika virus epidemic was about the risk that their pregnancy carried and what could happen to their baby. For the health professional, it was essential to define the profile of pregnancies that were at the most risk of presenting with adverse events, especially in their most severe form. Relatively few patients were lost to follow-up in this study, which could be attributed to pregnant women in the study area having considered diagnostic testing and care to be essential, due to fear of the ZIKV epidemic and microcephaly. This temptation may be attributable to the extent of interest in the epidemic (including in the media), which may have led individuals to seek out a diagnosis at a time when diagnostic testing and care were only being offered to pregnant women. Additionally, our group attempted to reduce losses to follow-up by contacting all the women, as data on births and fetal deaths in other municipalities were unavailable.

The clinical picture presented by pregnant women showed no signs of severity. There were no marked differences, despite confirming infection by Zika or by another infectious agent that causes a rash, as mentioned by authors in other regions in the Americas and other regions of Brazil [9,10,11,12]. In addition to the rash, pruritus, and arthralgia of small joints of the hands and feet, headache and non-secretive conjunctivitis were the symptoms most reported by the pregnant women included in the study.

There were no differences in social, demographic, or obstetric characteristics associated with the confirmation of Zika virus infection, except for contact with people with similar symptoms, suggesting that pregnant women with confirmed infection could have been under more intense exposure to vector transmission.

It was observed that the time interval between the onset of symptoms and the collection of biological samples for laboratory confirmation was shorter among those who were positive, which does not allow us to rule out that those pregnant women who did not have laboratory confirmed infection could be false negatives with a lower viral load. The proportion of suspected cases of maternal ZIKVi in pregnancy that tested negative in the current study was similar to that observed by Brasil et al. [4,6] in Rio de Janeiro, but was lower than that observed by Nogueira et al. [11] in São José do Rio Preto. This suggests that there are limits to the diagnostic ability of the test used to confirm cases. In a study of women from French territories in the Americas, Hoen et al. [12] also analyzed a pregnant cohort; however, their report did not describe the pregnancy outcomes in unconfirmed infections. Together with these prior studies, the results of our study raise the question, “If these symptomatic patients did not have ZIKV infection, then what did they have?” In the Rio de Janeiro cohort, 42% of women with negative RT-PCR ZIKV findings had confirmed Chikungunya infections. Neither the Sao José do Rio Preto nor the French Guiana study estimated the frequency of adverse outcomes in symptomatic pregnant women with negative ZIKV test findings. Further, neither of these studies examined the non-ZIKV causes of the symptoms that were presented by the patients.

Although it is not possible to say that pregnant women in the first trimester have a higher frequency of infection, when compared to pregnant women in other trimesters of pregnancy, our results suggest that all adverse effects observed in the fetus or child—especially the most severe—were more frequent in pregnant women who were symptomatic in the first trimester of pregnancy, especially those who had laboratory confirmed infection.

The adverse effects on the fetus and the child exposed to ZIKV during maternal infection in pregnancy may be directly due to vertical transmission or indirectly by the inflammatory process suffered by the infected pregnant woman or the infected placenta. In our study, there was no evidence of a relationship between fetal loss and laboratory confirmation of Zika virus infection, which suggests that any event that may be clinically similar to this infection may be equally harmful to the evolution of pregnancy. Spontaneous abortion was the most frequent form of fetal loss among pregnant women with confirmed infection, while stillbirth was more frequent among those who had no laboratory confirmed Zika virus infection. Our results suggest that maternal Zika virus infection is a risk factor for premature birth.

Microcephaly is, perhaps, the most feared outcome for pregnant women who have a rash disease, affecting a small proportion of infected pregnant women; presenting infection during the first trimester of pregnancy was a significant risk factor for microcephaly.

In this study, we evaluated pregnant women who had a rash disease during pregnancy by comparing those whose infection could be confirmed by RT-PCR with those in which this molecular examination was not positive. Due to the limitations of the study, it is possible that the pregnant women evaluated in the study were not part of different groups. Additionally, due to the design of the study, our results do not allow us to conclude whether the adverse effects observed in fetuses or newborns exposed to ZIKV during maternal infection were due to or mediated by the virus. The frequency of adverse pregnancy and child events found in this study was lower than that evidenced in studies carried out in other geographic areas [4,5,6,9,10,11,12,13]. This can be explained by the differences in the recruitment of research participants. In areas of vector transmission, the cases that can be diagnosed and better followed are symptomatic. When exposure is sporadic, as is possible through horizontal infections among partners of travelers, the diagnostic methods follow different flowcharts of application. Likewise, places where the problem was detected by the occurrence of adverse outcomes show variation in the association between the virus infection and its potential effects on the pregnant woman and the baby.

Finally, our results indicate the importance of monitoring symptomatic pregnant women, even when no infection is diagnosed during the acute event, as they are not exempt from presenting adverse pregnancy outcomes—including the most serious ones.

## 5. Conclusions

In conclusion, our results show that the frequency of severe adverse pregnancy outcomes was high among women infected with ZIKV during the period of peak ZIKV transmission in Manaus. The study findings indicate that maternal infection may be especially devastating to the fetus when symptoms are presented during the first trimester of pregnancy. In pregnant women with ZIKVi, the serious adverse effects seen in the fetus, such as microcephaly, miscarriage, or stillbirth, can be attributed to the direct action of the virus on the fetal tissue or indirectly by maternal infection. Finally, the difficulty in associating maternal ZIKV infection with adverse effects on the fetus, as observed in some cases in our study, may be due to the limitations of the diagnostic test itself.

## Figures and Tables

**Figure 1 viruses-12-01362-f001:**
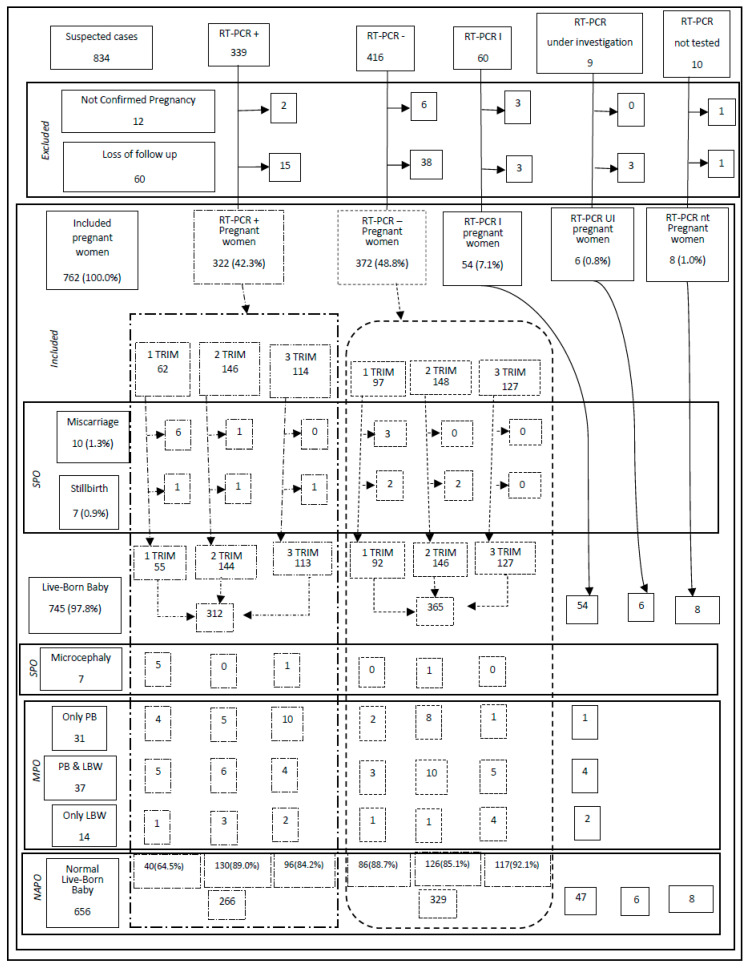
Recruitment, follow up, and pregnancy outcomes in women registered as having been exposed to Zika virus infection (ZIKVi) in pregnancy. The reported cases are from the Tropical Medicine Foundation Dr Heitor Vieira Dourado (FMT-HVD), Manaus, Brazil. RT-PCR: reverse transcription polymerase chain reaction; +: positive; −: negative; I: indeterminate; UI: under investigation; NT: not tested; PB: Premature birth; LBW: Low birth weight; PB & LBW: Low birth weight cases associated with premature birth; TRIM: trimester of pregnancy at the onset of symptoms; SOP: severe pregnancy outcome; MOP Moderate adverse outcomes in pregnancy; NAOP: No adverse pregnancy outcomes identified.

**Figure 2 viruses-12-01362-f002:**
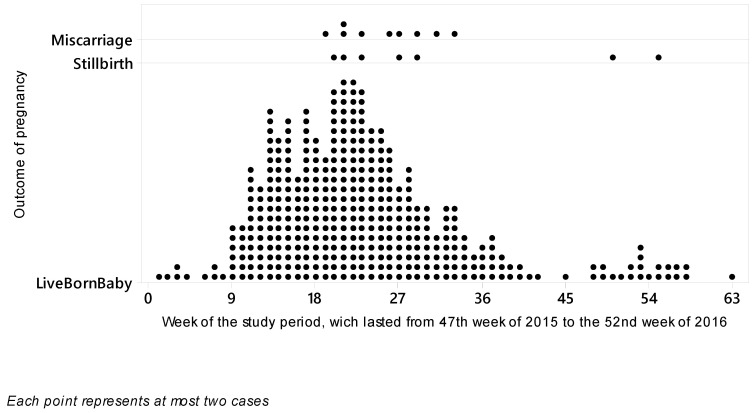
Distribution of registered cases of suspected or confirmed maternal ZIKVi in pregnancy, during the period lasting from the 47th epidemiological week of 2015 to the 52nd epidemiological week of 2016. Each point indicates the symptom onset time, stratified according to the gestational outcome (Miscarriage, Stillbirth, or live-born baby). The reported cases are from the Tropical Medicine Foundation Dr Heitor Vieira Dourado (FMT-HVD, Manaus, Brazil).

**Figure 3 viruses-12-01362-f003:**
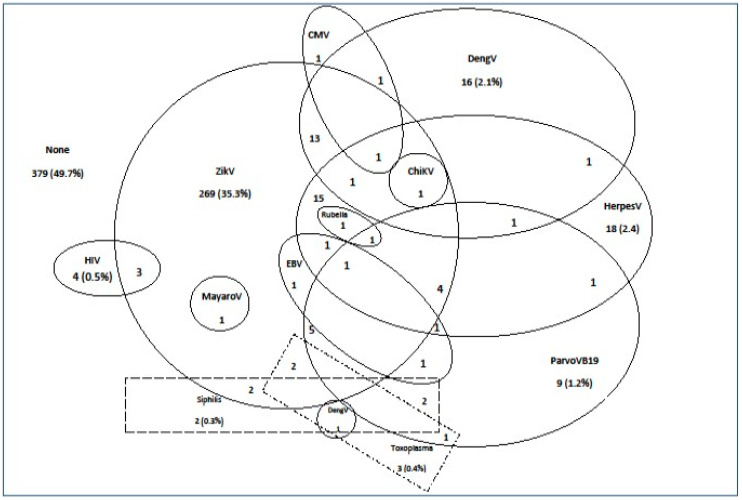
Distribution of serological markers of infections and coinfections in a cohort of women in Manaus who had exanthematic diseases registered as ZIKVi in pregnancy. ChikV: Chikungunya virus; CMV: cytomegalovirus; DengV: Dengue virus; EBV: Epstein–Barr virus; HerpesV: herpes virus; HIV: Human immunodeficiency virus; MayaroV: Mayaro virus; ParvoV: Parvovirus B19; ZIKV: Zika virus. None: no serological marker of infection found.

**Table 1 viruses-12-01362-t001:** Characteristics of a cohort of pregnant women with exanthematic disease during the period when Zika virus transmission was most intense in Manaus, according to the results of RT-PCR ZIKV testing and the end of pregnancy.

Characteristics of a Cohort of Pregnant Women	ZIKV +	ZIKV Not +	All No. (%)	RR	[CI 95%]	*p*
	**No. 337 (100.0%)**	**No. 485 (100.0%)**	**No. 822 (100.0)**			
Lost to follow-up before birth of infant	15/337 (4.5)	45/485 (9.3)	60/822 (7.3)	0.8	(0.5–0.2)	<0.001
Pregnancies followed until delivery	322/337 (95.5)	440/485 (90.7)	762/822 (92.7)			
Age Median ± SD (years)	27.0 ± 6.4	27.6 ± 6.4	27.3 ± 6.4			>0.05
Parity	2.4 ± 1.5	2.3 ± 1.4	2.4 ± 1.4			>0.05
First pregnancy	90 (28.0)	124 (28.2)	214 (28.1)			>0.05
Time of symptoms evolution (days)	2.7 ± 2.9	2.9 ± 4.3	3.4 ± 3.8			<0.001
Week of pregnancy at the onset of symptoms	22.7 ± 8.8	22.2 ± 9.8	22.4 ± 6.4			>0.05
infected in 1st Trim	62/322 (19.3)	112/440 (25.5)	174/762 (22.8)			
infected in 2nd trim	146/322 (45.3)	177/440 (40.2)	323/762 (42.4)			
infected in 3rd trim	114/322 (35.4)	151/440 (34.3)	265/762 (34.8)			
Fetal Loss	10/322 (3.1)	7/440 (1.6)	17/762 (2.2)	2.0 *	(0.8–5.1)	>0.05
in 1st trim	7/62 (11.3)	5/112 (4.5)	12/174 (6.9)	2.5 *	(0.8–7.6)	>0.05
in 2nd trim	2/146 (1.4)	2/177 (1.1)	4/323 (1.2)	1.2 *	(0.2–8.5)	>0.05
in 3rd trim	1/114 (0.9)	0/151 (0.0)	1/265 (0.4)			
Live births	312/322 (96.9)	433/440 (98.4)	745/762 (97.8)			
infected in 1st trim	55/62 (88.7)	107/112 (95.5)	162/174 (93.1)			
infected in 2nd trim	144/146 (98.6)	175/177 (98.9)	319/323 (98.8)			
infected in 3rd trim	113/114 (99.1)	151/151 (100.0)	264/265 (99.6)			

ZIKV +: RT-PCR positive for ZIKV; ZIKV not +: RT-PCR non-positive for ZIKV. * Compared to pregnancy with live birth per trimester of pregnancy.

**Table 2 viruses-12-01362-t002:** Summary of symptoms reported by pregnant women with exanthematic disease during the period when Zika virus transmission was most intense in Manaus, according to the diagnosed infections.

Variable	All	Single ZIKVi	Single DENVi	Coinfection ZIKV & DENV	ZIKVi & Other Infection	Other Infection	NONE
	**762 (100.0%)**	**269 (35.5%)**	**16 (2.1%)**	**16 (2.1%)**	**37 (4.9%)**	**45 (5.9%)**	**379 (49.7%)**
Rash	96.8	99.1	100.0	100.0	100.0	95.5	94.5
Pruritus	92.0	93.5	100.0	87.5	91.9	95.5	90.2
Lymphadenopathies	7.1	6.2	14.3	12.5	15.2	8.1	5.4
Contact	64.9	72.9	66.7	43.8	41.7	41.0	44.3
Hand arthralgia	61.2	39.0	44.4	62.5	58.3	41.0	61.2
Odynophagia	6.9	4.4	0.0	13.3	14.3	5.3	7.8
Headache	57.5	54.3	72.7	40.0	38.9	58.5	63.2
Foot arthralgia	54.0	56.0	33.3	68.8	55.6	33.3	55.7
Myalgia	52.5	49.4	55.6	68.8	42.9	45.0	56.5
Vaginal bleeding	5.4	5.9	0.0	6.3	3.0	2.6	6.2
Fever	49.8	47.0	66.7	43.8	32.4	47.7	54.2
Eye burning	48.0	50.3	70.0	56.3	40.0	42.5	48.0
Conjunctivitis	47.0	52.2	66.7	43.8	41.7	41.0	44.3
Hand edema	45.3	54.5	37.5	56.3	61.1	35.1	35.7
Foot edema	43.9	49.0	37.5	53.3	52.8	35.1	37.8
Asthenia adynamia	42.5	38.7	33.3	40.0	45.7	52.6	43.6
Vomiting	23.0	22.2	0.0	0.0	0.0	33.3	26.7
Uterine contractions	19.7	21.8	0.0	18.8	17.6	16.7	19.9
Diarrhea	18.5	19.6	12.5	18.8	17.7	20.5	17.5
Ocular pruritus	15.6	15.5	0.0	33.3	14.3	6.9	17.2

ZIKVi: maternal Zika virus infection; DengVi: maternal Dengue virus infection. Other maternal infections represent cases of maternal Chikungunya virus infection; maternal cytomegalovirus infection; maternal Epstein–Barr virus infection; maternal herpes virus infection; maternal Human immunodeficiency virus infection; maternal Mayaro virus infection; maternal Parvovirus B19 infection; None: no serological marker of infection found.

**Table 3 viruses-12-01362-t003:** Pregnancy outcomes in a cohort of exanthematic pregnant women during the period when Zika virus transmission was most intense in Manaus, according to the diagnosed infections.

Variable	Single ZIKVi	Single DENVi	Coinfection ZIKV & DENV	ZIKVi & Other Infection	Other Infection	NONE	All
	269 (35.5%)	16 (2.1%)	16 (2.1%)	37 (4.9%)	45 (5.9%)	379 (49.7%)	762 (100.0%)
Adverse pregnancy outcome	19.0	0.0	12.5	8.1	13.3	11.6	13.9
PTD and LBW	4.6	0.0	6.3	5.4	6.8	5.1	5.0
Premature delivery	7.3	0.0	0.0	0.0	2.3	3.2	4.3
Low birth weight	1.5	0.0	6.3	2.7	2.3	1.6	1.8
Miscarriage	2.6	0.0	0.0	0.0	2.2	0.5	1.3
Microcephaly	2.2	0.0	0.0	0.0	0.0	0.3	0.9
Stillbirth	1.1	0.0	0.0	0.0	0.0	1.1	0.9

ZIKVi: maternal Zika virus infection; DengVi: maternal Dengue virus infection. Other maternal infections represent cases of maternal Chikungunya virus infection; maternal cytomegalovirus infection; maternal Epstein–Barr virus infection; maternal herpes virus infection; maternal Human immunodeficiency virus infection; maternal Mayaro virus infection; maternal Parvovirus B19 infection; None: no serological marker of infection found.

**Table 4 viruses-12-01362-t004:** Pregnancy outcomes in a cohort of exanthematic pregnant women during the period when Zika virus transmission was most intense in Manaus, according to the RT-PCR for ZIKV infection.

Pregnancy Outcomes	ZIKV + No. (%)	ZIKV − No. (%)	All No. (%)	RR	[CI 95%]	*p*
	**No.322 (100.0%)**	**No. 440 (100.0%)**	**No. 762 (100.0)**			
Live births	312/322 (96.9)	433/440 (98.4)	745/762 (97.8)			
infected in 1st trim	55/62 (88.7)	107/112 (95.5)	162/174 (93.1)			
infected in 2nd trim	144/146 (98.6)	175/177 (98.9)	319/323 (98.8)			
infected in 3rd trim	113/114 (99.1)	151/151 (100.0)	264/265 (99.6)			
Male sex No. (%)	160 (51.3)	234 (54.0)	394 (52.9)			>0.05
Birth weight (g)	3248.8 ± 575.0	3292.9 ± 522.1	3274.5 ± 545.0			>0.05
Only PB	19/312 (6.1)	13/433 (3.0)	32/745 (4.3)	2.1 ^§^	(1.03–4.1)	<0.05
infected in 1st trim	4/55 (7.2)	2/107 (1.9)	6/162 (3.7)	4.6 ^§^	(0.9–28.4)	>0.05
infected in 2nd trim	5/144 (3.5)	9/175 (5.1)	14/319 (4.3)	0.7 ^§^	(0.2–2.0)	>0.05
infected in 3rd trim	10/113 (8.8)	2/151 (1.3)	12/264 (4.5)	6.6 ^§^	(1.4–29.5)	<0.01
Only LBW	6/312 (1.0)	7/433 (1.6)	13/745 (1.7)	1.3 ^§^	(0.4–3.7)	>0.05
infected in 1st trim	1/55 (1.8)	1/107 (0.9)	2/162 (1.2)	2.5 ^§^	(0.03–198.2)	>0.05
infected in 2nd trim	3/144 (2.1)	1/175 (0.6)	4/319 (1.3)	3.5 ^§^	(0.4–32.7)	>0.05
infected in 3rd trim	2/113 (1.8)	5/151 (3.3)	7/264 (2.7)	0.6 ^§^	(0.11–3.0)	>0.05
Both PB & LBW	15/312 (4.8)	22/433 (5.0)	37/745 (5.0)	1.0 ^§^	(0.5–1.9)	>0.05
infected in 1st trim	5/55 (9.1)	4/107 (3.7)	9/162 (5.6)	2.9 ^§^	(0.8–10.3)	>0.05
infected in 2nd trim	6/144 (4.2)	12/175 (6.9)	18/319 (5.6)	0.6 ^§^	(0.2–1.6)	>0.05
infected in 3rd trim	4/113 (3.5)	6/151 (4.0)	10/264 (3.8)	1.0 ^§^	(0.3–3.3)	>0.05
Microcephaly	6/312 (1.9)	1/433 (0.2)	7/745 (0.9)	8.6 ^§^	(1.04–71.2)	<0.05
infected in 1st trim	5/55 (9.1)	0/151 (0.0)	5/162 (3.1)			
infected in 2nd trim	0/144 (0.0)	1/175 (0.6)	1/319 (0.3)			
infected in 3rd trim	1/113 (0.9)	0/151 (0.0)	1/264 (0.4)			
Any Adverse pregnancy outcome	56/322 (17.4)	50/440 (11.4)	106/762 (13.9)	1.6 *	(1.1–2.5)	<0.05
infected in 1st trim	22/62 (35.4)	12/112 (10.7)	34/174 (19.5)	3.3 *	(1.7–6.2)	<0.001
infected in 2nd trim	16/146 (11.0)	25/177 (14.1)	41/323 (12.7)	0.8 *	(0.4–1.4)	>0.05
infected in 3rd trim	18/114 (15.8)	13/151 (8.6)	31/265 (11.7)	1.8 *	(0.9–3.6)	>0.05
Severe Pregnancy outcome	16/322 (5.0)	8/440 (1.8)	24/762 (3.1)	3.1 *	(1.4–7.3)	<0.01
infected in 1st trim	12/62	5/112	17/174	4.8 *	(1.8–13.0)	<0.005
infected in 2nd trim	2/146	3/177	7/323	0.8 *	(0.1–4.6)	>0.05
infected in 3rd trim	2/114	0/151	2/265	-	-	-
Moderate Pregnancy outcome	40/322 (12.4)	42/440 (9.6)	82/762 (10.8)	1.3 *	(0.9–2.0)	>0.05
infected in 1st trim	10/62	7/112	17/174	3.1 *	(1.2–7.5)	<0.05
infected in 2nd trim	14/146	22/177	36/323	0.8 *	(0.4–1.4)	>0.05
infected in 3rd trim	16/114	13/151	29/265	1.7 *	(0.8–3.3)	>0.05
Healthy new born babies	266/322 (82.6)	390/440 (88.6)	656/762 (86.1)			
infected in 1st trim	40/62 (64.5)	100/112 (89.3)	140/174 (80.5)			
infected in 2nd trim	130/146 (89.0)	152/177 (85.9)	282/323 (88.4)			
infected in 3rd Trim	96/114 (84.2)	138/151 (91.4)	234/265 (88.6)			

ZIKV +: RT-PCR positive for ZIKV; ZIKV not +: RT-PCR non-positive for ZIKV; PB: premature birth without LBW; LBW: low birth weight without premature birth; PB & LBW: Low birth weight due to premature birth. * Compared to pregnancy with live birth per trimester of pregnancy. ^§^ Compared to pregnancy with live birth without adverse outcomes per trimester of pregnancy.

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
