# Peer review of "Zika Virus Infection in a Cohort of Pregnant Women with Exanthematic Disease in Manaus, Brazilian Amazon"

_viruses, 2020, doi:10.3390/v12121362_

Round 1
Reviewer 1 Report
Overview
This paper describes a relatively large cohort of pregnant women with exanthematic disease in Brazil, who were suspected to have Zika virus (ZIKV) infection, and referred to a specialist centre for infectious disease. Women were tested for ZIKV with RT-PCR, and also for other TORCH infections. The main tables presented divide women into the categories: ZIKV RT-PCR+ve (n=339), RT-PCR-ve (n=416), and indeterminate, unanalysed, not tested (n=79). Most of the results reported relate to maternal symptoms and maternal demographics. The authors report 4.96% “severe” outcomes in the ZIKV+ve group, and 2.15% in the ZIKV-ve group.
General comments
This is one of the largest studies so far, so publication will be welcomed. However, the way the results are presented could be greatly improved, and there should be a greater level of detail on pregnancy and newborn outcomes, and less detail on maternal symptoms and demographics. A distinct limitation of the study, which should be mentioned in discussion, is the complete lack of newborn testing for congenital infections, whether ZIKV or the classic TORCH infections, even in infants with severe outcomes. One would have thought this would be required for appropriate clinical management. This is in spite of the considerable focus on laboratory diagnosis of infections in pregnancy – not only arboviruses, which are known to be associated with exanthematic disease – but also parvovirus, toxoplasmosis, CMV, HIV, rubella … etc. It would be useful if the authors could clarify how their testing protocol relates to Brazilian national protocols for management of children with possible congenital infection. Or whether this data, perhaps in conjunction with pediatric clinical follow-up, will be the subject of a further paper.
Whether or not such laboratory testing has been done, if there is to be a follow-up paper, I would urge the authors to re-think their publication strategy, with one paper on symptoms in the mother, and another on pregnancy outcomes and newborn outcomes at least to one to three months.
As it stands the discussion section of the paper focuses almost entirely on the findings at delivery, while the results focus almost entirely on findings in the mothers. Authors need to decide what this paper is about.
Specific comments
- Abstract line 34: 834 women with suspected ZIKV, of whom 91.4% had confirmed pregnancy. This accords with Figure 2, but this figure is not picked up in the opening sections of the results. The effective study denominator is the 760 pregnant women. Lines 132-144 should be rewritten to clarify
- Abstract line 38: “4.96%” …. Authors should adjust all the reported percentages in text and in tables, to one decimal place.
- Definitions line 81. Authors have grouped together, as “severe”, a number of disparate outcomes including abortion (pregnancy interrupted <22 weeks), stillbirth and microcephaly (<2 SD). In a paper reporting outcomes of pregnancy, we would expect neurological and ophthalmological examinations, limb abnormalities, and so one. Were these not done, or are they to be reported elsewhere. They should be reported here.
- Line 84. “A case was defined … “ Make it clear that this section is about maternal infection, not congenital infection or newborn findings.
- Section 3.2 and the “Outcome of pregnancy figure”. Could this be redrawn as a bar chart with numbers on the left hand Y-axis? Is this the 834, or the 1286 “suspected cases”?
- Figure 2. This is in some ways an admirably clear account of the cohort. Are the micro-cephaly, PTD, LBW, and Normal live baby categories mutually exclusive ? Is there a reason why the “Normal” babies are not split into the three trimesters? – it
- Table 1: I wonder the maternal age, marital status, and schooling needs to be broken down by RT-PCR category. It could just be reported that there was no difference. In addition, “widowed” could be combined with “separated/divorced”. Is “ignored” the same as “not known”?
The one part of the table where the breakdown by RT-PCR result is of interest is the trimester breakdown. The authors should include the % here. This is an important table as differences in the trimester distribution by PCR result can point to potential biases in ascertainment, such as selective recruitment of severe cases. This will deserve comment in a revised draft, it should be a separate table.
The Outcomes of pregnancy should be a separate table and reported by trimester and in greater detail. I would suggest that they are reported by trimester of maternal infection AND by RT-PCR status, and that “abortions”, stillbirths, and live births are distinguished, and within each of these three categories, the clinical findings (microcephaly / other neurological and ocular abnormalities are set out – as much as this is possible). Would any cases qualify as “Congenital Zika Syndrome”? This should be reported based on the same definitions that have become standard, or something similar (for example Hoen). Were there any cases of microcephaly <3SD? Were there any cases of cranial disproportion ? These kinds of issues should be the focus of the results.
A specific point about “abortions” this term tends to refer to terminations of pregnancy in English and American usage. Would it be more correct to refer to “aborted fetus” or even “fetal loss”?. Further: is it possible to distinguish between terminations and other fetal loss?
A key point is that the “headline” figures seen in the abstract (4.96%) and (2.15%) do not appear anywhere in the results section. There is a reported 10.5 OR for 1st Trimester severe outcomes vs trimesters 2&3. Is this just in the PCR +ve group? Are the ORs the same in PCR+ve and PCR-ve ?
The section on “clinical condition” again does not look too interesting: I see little to distinguish RT-PCR +ve and -ve cases – suggesting the PCR -ves probably had ZIKV.
- Table 2. The level of detail is again somewhat poorly judged, with so little data on Toxo, HIV, CMV and so on. Perhaps the headings could be: ZIKV (+/- other infections), DENV (+/- others), Both ZIKV and DENV, Other infections, No Infections. This might, or might not supply evidence about whether the PCR-ve were ZIKV or not
- Authors should re-arrange the results: first the findings on the mothers; then the outcomes of pregnancy
- Once again …. One decimal place for percentages is adequate: two makes it harder to read.
- Line 189: the heading “Torch Syndrome” will be very confusing for readers, who main focus will be on newborn outcomes: the results discussed in this section are not TORCH syndrome, but findings in the mother.
- Lines 203-213. This should be rewritten as a discussion around the revised section of Table 1 relating to outcomes.
- Table 3: I do not think it is useful to have a table suggesting a higher risk of severe outcomes in single, basic education and first pregnancy. It is difficult to interpret, as the baseline group to which the ORs are relative must be different for “Single”, “Basic education”, first pregnancy, first trimester, RT-PCR+; and it is not entirely obvious what the baseline category is. It cannot be “teenager”.
- The discussion should include a frank descriptions of study limitations, alongside its good points. On the good side, the authors have a relatively large cohort and have included follow-up of “RT-PCR -ve” cases. On the bad side, there was a dearth of laboratory follow-up of newborns, even those with “severe” Inclusion of only symptomatic women is another limitation, lack of clinical follow-up of infants, and poor level of detail on the newborn examination (unless this can be addressed in revision).
Possibly some of the limitations are attributable to the Brazilian national protocols for surveillance of ZIKV in pregnancy and care of the mother and child in the context of ZIKV
- The discussion should refer to the papers by Pomar, and authors should compare the rates of adverse outcomes in previous papers to what they find in their cohort. My impression is that the rates of severe outcomes – if that is to include stillbirths and aborted fetuses – is lower than other studies.
At a more fundamental level, if the paper is to focus its results section so much on maternal symptoms, the discussion needs to refer in much more detail to a comparison with maternal findings in previous literature
- The text should be reviewed for English idiom as there are a few phrases that are not quite right: such as line 237 “does not allow to know” …
Reviewer 2 Report
This is important work since there are still questions not answer about Zika related adverse pregnancy outcomes. However, there are essential concerns to be addressed in this manuscript. First, in the introduction, the research question is not clearly described, and the authors did not present a critical evaluation of the existent literature. The authors just numerate potential gaps in the literature but do not clearly say which gaps this paper will be filled.
In the methods section, you stated that any interruptions occurred from 22-37 weeks was recorded as preterm birth. What about stillbirths? You must differentiate if they were born alive (preterm) or not (stillbirth). Under statistical analyses, you said that you used z-score, but z-score is not a test. You must say which statistical test you used, how the variables were summarized (means, medium). I would encourage to include an analysis of women with confirmed ZIKA and symptoms in the first trimester.
In the results section, some numbers are not the same in the main text and tables, for example, loss of follow up was 60 in the main text and 63 in the table. In the first paragraph of the results, you used data from the surveillance and SINASC, but you did not include this information in the methods. I would encourage you to drop the first paragraph of results and state that you included xx women and this number corresponded to xx% of the total number of notified cases of suspected Zika among pregnant women in the state of Manaus. I also would encourage you to include a flow diagram. I did not understand how you cannot confirm pregnancy in a cohort of pregnant women (this is not clear for me). The paragraph 186-188 should be in the methods. What is as soon as possible? The patients were not tested when they were enrolled in the cohort when they had symptoms?
The discussion should be re-written, and in the first paragraph, you should include only the summary of your findings, the discussion-based in the literature and include a paragraph only with the strengths and limitations of the paper and how it could bias your results.
Round 2
Reviewer 1 Report
The authors have effected many changes that have greatly improved the depth of information provided by their paper
I have some remaining concerns, all of which have to do with wording.
1.. there are a number of incorrect idioms and usages, such as "no tested" instead of "not tested", "moderated" instead of "moderate". I recommend a native speaker of English is asked to check the text thorughly
2.. In Fig 1, there is some inconsistency in that some boxes have numbers by themselves, others "N" followed by a number, others "N." followed by a number. A minor problem of course
3... But here is a major issue: throughout the text there is reference to "ZIKV infection", or "infection", but it is not exactly clear whether authors are referring to maternal infection or fetal/congenital infection. To readers like myself, who have studied congenital infection for many years, some of the text is baffling.
examples:
Line 309: "Our results show that ZIKV infection is a risk factor for premature birth ...". I think it means maternal. But whether this is due to, or mediated by fetal infection is also an issue.
Line 334: " In pregnant women with suspected ZIKV in pregnancy, most cases of adverse pregnancy outcomes can be attributed to ZIKV infection (in the baby or the mother?) "
I suggest authors go through the text very carefully, reviewing this aspect of the wording.
4.. I think the lack of clarity on this point has resulted in some quite strange comments: Line 301: "Although it is not possible to state that pregnant women in the first trimester are more susceptible to ZIKV infection, our results show that all adverse outcomes, especially the most serious ones, were more frequent in pregnant women who were symptomatic in the first trimester ... especially those who had lab confirmed ZIKV".
Surely you are not suggesting that women are more likely to be infected in T1 than in T2 or T3???
Or do you mean that the fetus is more susceptible in T1 ?
5.. While outcomes like microcephaly are clear the result of transmission of ZIKV in utero, the authors might take the opportunity to speculate on whether the increased levels of LBW, prematurity, fetal loss, and stillbirth are a result of fetal infection, or a result of maternal infection in the absence of transmission to the fetus.
6. The authors should be congratulated on an interesting paper
Author Response
The authors have effected many changes that have greatly improved the depth of information provided by their paper
I have some remaining concerns, all of which have to do with wording.
1.. there are a number of incorrect idioms and usages, such as "no tested" instead of "not tested", "moderated" instead of "moderate". I recommend a native speaker of English is asked to check the text thoroughly
Response: We identified the errors pointed out by the reviewer and corrected them both in the text and in the figures. The manuscript has been revised in English. If the reviewers consider it necessary, the authors agree that the manuscript goes through the English edition service of MDPI.
2.. In Fig 1, there is some inconsistency in that some boxes have numbers by themselves, others "N" followed by a number, others "N." followed by a number. A minor problem of course
Response: We identified the inadequacy pointed out by the reviewer and standardized all the number boxes in figure 1.
3... But here is a major issue: throughout the text there is reference to "ZIKV infection", or "infection", but it is not exactly clear whether authors are referring to maternal infection or fetal/congenital infection. To readers like myself, who have studied congenital infection for many years, some of the text is baffling.
Response: We agreed with the reviewer that the text as it was written could cause confusion in the reader of Viruses and decided to add the word "maternal" to ZIKVi when it was necessary to specify that we were talking about maternal infection as a risk factor for the child.
examples:
Line 309: "Our results show that ZIKV infection is a risk factor for premature birth ...". I think it means maternal. But whether this is due to, or mediated by fetal infection is also an issue
Response: The sentence pointed out by the reviewer was written again as follows: "Our results suggest that maternal Zika virus infection is a risk factor for premature birth" lines 324 - 325. The following text was added to the study's limitations: Additionally, due to the design of the study, our results do not allow us to conclude whether the adverse effects observed in fetuses or newborns exposed to ZIKV during maternal infection are due to or mediated by the virus. Lines 332 – 334.
Line 334: " In pregnant women with suspected ZIKV in pregnancy, most cases of adverse pregnancy outcomes can be attributed to ZIKV infection (in the baby or the mother?) " I suggest authors go through the text very carefully, reviewing this aspect of the wording.
Response: The following text was added on lines 332 to 334: “The adverse effects on the fetus and the child exposed to ZIKV during maternal infection in pregnancy can be directly due to vertical transmission or indirectly by the inflammatory process suffered by the infected pregnant woman or the infected placenta” The sentence cited by the reviewer was replaced by this one on lines 350 to 354: "In pregnant women with ZIKVi, the serious adverse effects seen in the fetus, such as microcephaly, miscarriage, or stillbirth, can be attributed to the direct action of the virus on the fetal tissue or indirectly by maternal infection. Finally, the difficulty in associating maternal ZIKV infection with adverse effects on the fetus, as observed in some cases in our study, may be due to the limitations of the diagnostic test itself”.
4.. I think the lack of clarity on this point has resulted in some quite strange comments: Line 301: "Although it is not possible to state that pregnant women in the first trimester are more susceptible to ZIKV infection, our results show that all adverse outcomes, especially the most serious ones, were more frequent in pregnant women who were symptomatic in the first trimester ... especially those who had lab confirmed ZIKV".
Response: The sentence cited by the reviewer was edited in lines 312 to 316 as follows: "Although it is not possible to say that pregnant women in the first trimester have a higher frequency of infection when compared to pregnant women in other trimesters of pregnancy, our results suggest that all adverse effects observed in the fetus or child, especially the most severe, were more frequent in symptomatic pregnant women in the first trimester of pregnancy, especially those who had laboratory confirmed infection. "
Surely you are not suggesting that women are more likely to be infected in T1 than in T2 or T3???
Response: I mean that I cannot say that the pregnant woman in the first trimester of pregnancy is more susceptible to infection because I would need to have a control group of exposed uninfected pregnant women and compare the proportion of infected women per trimester of pregnancy.
Or do you mean that the fetus is more susceptible in T1 ?
Response: Yes. This interpretation of our text is the correct one.
5.. While outcomes like microcephaly are clear the result of transmission of ZIKV in utero, the authors might take the opportunity to speculate on whether the increased levels of LBW, prematurity, fetal loss, and stillbirth are a result of fetal infection, or a result of maternal infection in the absence of transmission to the fetus.
Response: We followed the reviewer's suggestion and added the following text to the manuscript (lines 350 to 352): “In pregnant women with ZIKVi, the serious adverse effects seen in the fetus, such as microcephaly, miscarriage, or stillbirth, can be attributed to the direct action of the virus on the fetal tissue or indirectly by maternal infection”.
6. The authors should be congratulated on an interesting paper
Response: The authors are very grateful for all contributions made by the reviewers to our manuscript. It is the result of hard work by a highly committed multidisciplinary team.

Reviewer 2 Report
I think the discussion could be improved. The first paragraph should summarize the findings of the study and should be improved.
The limitations and strengths of the study should also be emphasized.
Author Response
I think the discussion could be improved. The first paragraph should summarize the findings of the study and should be improved.
Response: The entire text has been revised to make it clearer and more objective. In addition, following the reviewer's suggestion, the following text was included in the first paragraph of the discussion (lines 266 to 275):”This study shows the evolution of pregnancy, from the onset of symptoms to the end of pregnancy, in a large population of pregnant women with rash disease during a period of intense local transmission of the Zika virus in the Brazilian Amazon. Although less than half of the pregnant women had their infection confirmed in the laboratory, adverse events in the fetus and child were observed in the entire population studied. Other diseases that cause similar conditions in pregnant women and fetal infections by vertical transmission have also been studied and have not been identified in cases with severe adverse pregnancy outcome except in a fetal loss associated with Toxoplasmosis and Parvovirus B19-coinfection. Serious adverse outcomes in the fetus, such as miscarriage, stillbirth, or microcephaly, were more likely to occur in pregnant women who had confirmed Zika virus infection especially if symptoms started in the first trimester of pregnancy”.
The limitations and strengths of the study should also be emphasized
Response: Also following the reviewer's suggestion, the study's limitations were modified and now the following text appears in lines 329 to 334 of the manuscript: “This study evaluated pregnant women who had the rash disease during pregnancy by comparing those whose infection can be confirmed by RT-PCR with those in which this molecular examination was not positive. Due to the study’s limitations, it is possible that the pregnant women evaluated in the study are not part of different groups. Additionally, due to the design of the study, our results do not allow us to conclude whether the adverse effects observed in fetuses or newborns exposed to ZIKV during maternal infection are due to or mediated by the virus.”
The strengths of the study are also addressed in the first paragraph of the discussion when it shows that this study was carried out and a significant population in the Brazilian Amazon, looking for various causes of exanthematic disease that can produce potentially serious vertical transmission in the fetus and that followed until the end of pregnancy not only to pregnant women with laboratory confirmed Zika virus infection but all those who were symptomatic showing the importance of not excluding these pregnant women in the different studies.
